# SHOW, DON'T TELL: UNCOVERING IMPLICIT CHARACTER PORTRAYAL USING LLMS

## ABSTRACT

Tools for analyzing character portrayal in fiction are valuable for writers and literary scholars in developing and interpreting compelling stories. Existing tools, such as visualization tools for analyzing fictional characters, primarily rely on explicit textual indicators of character attributes. However, portrayal is often implicit, revealed through actions and behaviors rather than explicit statements. We address this gap by leveraging large language models (LLMs) to uncover implicit character portrayals. We start by generating a dataset for this task with greater cross-topic similarity, lexical diversity, and narrative lengths than existing narrative text corpora such as TinyStories and WritingPrompts. We then introduce LIIPA (LLMs for Inferring Implicit Portrayal for Character Analysis), a framework for prompting LLMs to uncover character portrayals. LIIPA can be configured to use various types of intermediate computation (character attribute word lists, chain-of-thought) to infer how fictional characters are portrayed in the source text. We find that LIIPA outperforms existing approaches, and is more robust to increasing character counts (number of unique persons depicted) due to its ability to utilize full narrative context. Lastly, we investigate the sensitivity of portrayal estimates to character demographics, identifying a fairness-accuracy tradeoff among methods in our LIIPA framework – a phenomenon familiar within the algorithmic fairness literature. Despite this tradeoff, all LIIPA variants consistently outperform non-LLM baselines in both fairness and accuracy. Our work demonstrates the potential benefits of using LLMs to analyze complex characters and to better understand how implicit portrayal biases may manifest in narrative texts.

## 1 INTRODUCTION

Computational tools for analyzing character portrayal in narratives facilitate bias detection in literary fiction (Lucy & Bamman, 2021; Fast et al., 2016) and AI-generated narratives (Huang et al., 2021). They also assist writers and literary scholars in refining their story drafts and character analyses (Hoque et al., 2023). Most of these existing tools rely on using *explicit* indicators in the text to uncover how a character is portrayed. However, portrayal is usually *implicit*, where a character's traits should be clear from their actions and behaviours rather than explicitly stated in the text (Chekhov & Yarmolinsky, 1954). For instance, "She was stranded on an island and built a boat to escape", which implicitly suggests high intelligence and resourcefulness. Uncovering implicit portrayal is more challenging than explicit portrayal, as it requires using commonsense knowledge to make inferences about how a character is portrayed. It becomes even more difficult to uncover from longer narratives depicting multiple distinct characters. Moreover, implicit portrayal is a key principle of literary design (Rimmon-Kenan, 2002) so it is concerning that most existing tools focus on visualizing characters using explicit indicators of portrayal. Furthermore, the evaluation of new methods for implicit portrayal is difficult due to the reliance of existing benchmarks on explicit character behavior to derive target labels (Mostafazadeh et al., 2016).

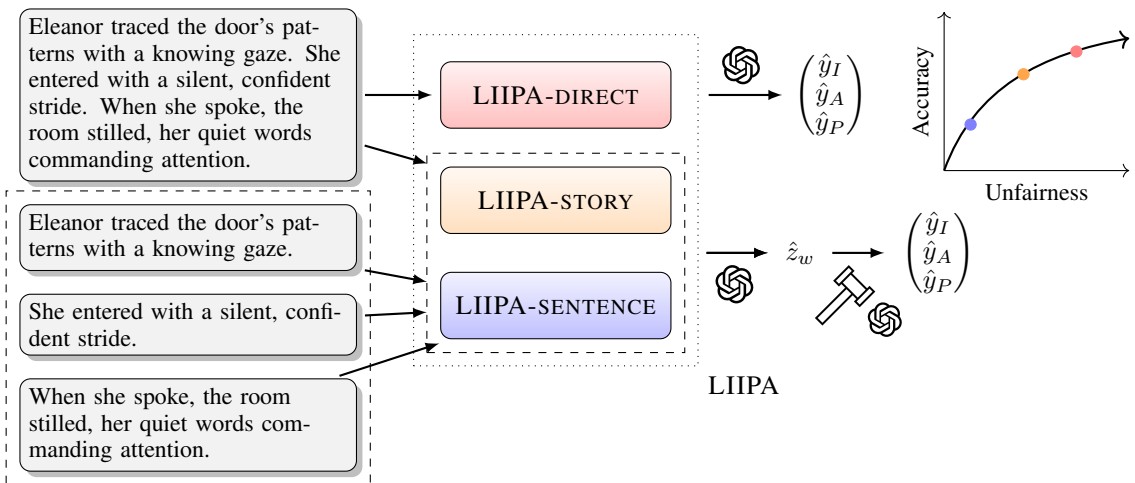

Figure 1: **LLMs for Inferring Implicit Portrayal for Character Analysis (LIIPA)**: Our proposed framework for uncovering implicit character portrayal using LLM prompting. LIIPA-DIRECT directly prompts an LLM to infer portrayal while LIIPA-STORY and LIIPA-SENTENCE generate intermediate word lists ($\hat{z}_w$) which are then mapped to portrayal labels using a separate LLM-based evaluator. We use intellect, appearance, and power as dimensions for portrayal. Each method represents a point on the fairness-accuracy Pareto frontier: LIIPA-DIRECT achieves the highest accuracy but with the least fairness, whereas LIIPA-SENTENCE minimizes unfairness but is less accurate.

Prior approaches to implicit character portrayal use the "Commonsense Transformer" (COMET) (Bosselut et al., 2019), a generative model for knowledge bases over text, to infer the mental state and motivations of protagonists (Huang et al., 2021; 2024). These methods are constrained by COMET's limitations in that it can only process simple event structures and cannot utilize long context lengths.

In this paper, we use LLMs to build expressive models for uncovering implicit character portrayal. We start by generating the first benchmark dataset designed specifically for this task. Compared with existing narrative text corpora such as TinyStories and WritingPrompts, our dataset offers greater cross-topic similarity, greater lexical diversity, and a broader representation of character roles. We then introduce a family of LLM prompting techniques that outperform the COMET-based approach for implicit character portrayal. We explore different prompting design choices, including the use of an intermediate character attribute list (as in COMET) to describe characters. We find that LLMs are more performant than the previous approach, although we also identify a fairness-accuracy tradeoff within LLM-based approaches. This suggests that care (beyond picking the "optimal" prompt) is required when designing socially beneficial tools for literary analysis.

Our contributions can be summarized as follows:

- We introduce **ImPortPrompts (Implicit Portrayal Prompts)**, a dataset for implicit character portrayal analysis with better diversity and coverage than existing benchmarks.

- We propose **LIIPA (LLMs for Inferring Implicit Portrayal for character Analysis)**, a framework for prompting LLMs to uncover character portrayals that outperforms COMET-based approaches. LIIPA has three variants: LIIPA-SENTENCE, LIIPA-STORY, and LIIPA-DIRECT each corresponding to a different form of intermediate computation (e.g. character attribute wordlist, chain of thought) used for estimating portrayal.

- We investigate fairness implications of this approach, and identify that various configurations (structured output, prompt types) of LIIPA realize a fairness-accuracy tradeoff (with LIIPA-DIRECT maximizing accuracy and LIIPA-SENTENCE minimizing unfairness). LIIPA-SENTENCE and LIIPA-STORY outperform the prior COMET baseline in both accuracy and fairness.

## 2 CURATING A NARRATIVE TEXT DATASET USING SYNTHETIC DATA GENERATION

### 2.1 TASK FORMULATION

We formulate the task of uncovering character portrayal from text as a multi-label classification problem. Our objective is to develop a function that maps an input narrative text and a specific character[1] from that text to a set of labels across three dimensions: intellect, appearance, and power (Figure 2). Each dimension is classified as either low, neutral, or high, with the "neutral" label reserved for cases where the text provides insufficient information to make a definitive inference about the character's portrayal. Formally, we aim to learn a function $f : \mathcal{X} \times \mathcal{C} \to \mathcal{Y}$ that takes a narrative text $x^{(i)} \in \mathcal{X}$ and a character $c_j^{(i)} \in \mathcal{C}$ from that text, and produces a set of labels $y_j^{(i)} \in \mathcal{Y}$ representing the character's portrayal. The label space $\mathcal{Y}$ is defined as the Cartesian product of the label spaces for each dimension: $\mathcal{Y} = \mathcal{Y}_{\text{intellect}} \times \mathcal{Y}_{\text{appearance}} \times \mathcal{Y}_{\text{power}}$, where each dimension's label space consists of the values {low, neutral, high}.

**Input**

Sarah's fingers flew across the keyboard, her eyes darting between multiple screens. She muttered complex algorithms under her breath. Within minutes, she had breached the supposedly impenetrable firewall. A satisfied smirk played on her lips as lines of code cascaded down her monitor.

**Output**

⚡ **Intellect:** High
👤 **Appearance:** Neutral
🛡 **Power:** Neutral

Figure 2: **Input-Output example for the character portrayal classification task.** The goal is to classify the character's intellect, appearance, and power (IAP) into {low, neutral, high} from an input narrative.

We select intellect, appearance, and power as our dimensions for character portrayal, as these have been previously studied in relation to social biases when analysing AI-generated narratives (Lucy & Bamman, 2021) and in comparison to human-written narratives (Huang et al., 2021; 2024). Our methods, however, can be readily adapted to other aspects of character portrayal such as emotional depth and moral alignment. We define "intellect" synonymously with logical-mathematical intelligence, as described by Patanella et al. (2011): *"the ability to think conceptually and abstractly, and the capacity to discern logical and numerical patterns."* Our definitions for appearance and power are less ambiguous and are detailed in Table 2 (§A.1), along with classification guidelines for what constitutes low, neutral, or high portrayal. Throughout this work, we abbreviate intellect, appearance, and power as IAP.

Our classification guidelines account for dynamic character portrayal, allowing for fluctuations in a character's attributes throughout the narrative. For instance, a protagonist who evolves from showing low to high intellect would be classified as exhibiting high intelligence overall. Unlike prior works (Huang et al., 2021; Brahman & Chaturvedi, 2020) that focus solely on protagonists, our formulation allows for the analysis of *any* character within the narrative. Note that our formulation here is designed to be flexible and gener-

---

[1]"Character" refers to a person depicted in the narrative, not a single letter/symbol used to compose the text sequence.

alizable, not constrained to either explicit or implicit character portrayal. The focus on *implicit* character portrayal can be implemented through careful constraints on the data generation, as we describe next.

## 2.2 DATASET CURATION METHODOLOGY

To curate a dataset for the implicit character portrayal classification task, we use LLMs to generate narrative texts under a set of controlled conditions such as character count (number of unique persons depicted) and narrative length. This approach allows us to increase the diversity of synthetically generated narratives while reducing representational biases that pertain to the controlled conditions (Yu et al., 2023). We formulate this process as generating a narrative text $x^{(i)}$ subject to a set of natural language constraints $C_i$. Below, we describe our constraints which are added as natural language instructions to the LLM prompt. A full, detailed list can be found in §A.3. We refer to this dataset as **ImPortPrompts (Implicit Portrayal Prompts)**.

**Natural Language Constraints:** The narrative must contain exactly $N_c^{(i)}$ characters and have a length of $L^{(i)}$ sentences. Each character should be assigned a role from {protagonist, antagonist, victim} and be given a character portrayal label set $y_j^{(i)} \in \mathcal{Y}$. To ensure implicit portrayal, each character must be portrayed implicitly through their actions, decisions, and interactions, rather than through explicit words and statements. The narrative should avoid using words that directly describe a character's intellect (e.g., intelligent, stupid, clever), appearance (e.g., beautiful, ugly, attractive), or power (e.g., powerful, weak, strong). The socio-demographic background of characters should not be stated or implied. This includes using gender-neutral names, avoiding mentions of racial characteristics, religious affiliations, and socioeconomic status. References to age, physical attributes, or cultural backgrounds that might reveal demographic information should also be omitted. The narrative genre and topic are selected from Table 4 (§A.4).

Our choice of character roles are motivated by prior works that develop automated methods for character role detection (Gomez-Zara et al., 2018) and extraction (Stammbach et al., 2022) in narratives. Our definitions for these roles can be found in Table 3 (§A.1) which allow for multiple protagonists and antagonists. Our socio-demographic constraint facilitates the measurement of fairness by eliminating socio-demographic information from the generated narratives as will be discussed in Section 3.2. The socio-demographic groups we use come from Gupta et al. (2024) which we repeat verbatim in Table 18 (§C.2).

**Experimental Setup:** We follow a similar setup to Perez et al. (2023) and use LLMs as narrative text generators that generate $x^{(i)}$ subject to constraints $C_i$. We can describe the ImPortPrompts generation process as sampling from a model subject to constraints $C_i$: $x^{(i)} \mid C_i \sim p_g(\cdot|C_i)$, where $p_g$ refers to the model we use to generate narratives, and $C_i = (y^{(i)}, C_i')$ with $C_i'$ representing the remaining constraints apart from the label set constraint. Our choices of $p_g$ are the GPT (OpenAI, 2024) and Claude (Anthropic, 2024) LLM families. We apply the tree-of-thoughts (ToT) prompting strategy to creative writing as done by Yao et al. (2023) to generate narratives that both more coherent and more likely to satisfy the constraints (prompts in § A.5). We also condition the narrative generation on a randomly sampled (genre, title) tuple from Table 4 (§A.4) which increases narrative diversity and helps to eliminate representation bias of topics in our data set. For quality assurance, we perform automated and human checks to ensure the LLM-generated narratives satisfy the natural language constraints outlined earlier, filtering out any narratives that fail our checks. Details of our data validation process can be found in §A.6.

## 2.3 EXPLORATORY DATA ANALYSIS

Next we explore the diversity and distributional properties of ImPortPrompts, while comparing it with existing narrative text corpora.

**Metrics:** We measure narrative quality using lexical and semantic diversity. For lexical diversity, we use the following indices: HD-D (Hypergeometric Distribution Diversity), Maas, and MTLD (Measure of Textual

Table 1: **Lexical and semantic diversity across datasets.** ↑ denotes that higher values of the metric indicate greater diversity, while ↓ signifies that lower values correspond to increased diversity. Lexical diversity metrics: HD-D (Hypergeometric Distribution Diversity), Maas (length-adjusted Type-Token Ratio), and MTLD (Measure of Textual Lexical Diversity). Semantic diversity metrics: Intra-topic APS (Average Pairwise Similarity), Inter-topic APS, and INGF (Inter-sample N-gram Frequency). Higher MTLD reflects more consistent lexical diversity across text lengths, and higher Inter-topic APS suggests more semantic similarity across topics.

| Dataset | Lexical Diversity | | | Semantic Diversity | | |
|---|---|---|---|---|---|---|
| | HD-D ↑ | Maas ↓ | MTLD ↑ | Intra-topic APS ↓ | Inter-topic APS ↓ | INGF ↓ |
| ImPortPrompts (Ours) | 0.77 | 0.02 | 67.39 | 0.85 | 0.49 | 0.04 |
| ROCStories | 0.73 | 0.02 | 45.78 | 0.82 | 0.14 | 0.01 |
| WritingPrompts | 0.77 | 0.02 | 47.66 | 0.80 | 0.26 | 0.02 |
| TinyStories | 0.73 | 0.03 | 44.95 | 0.84 | 0.45 | 0.02 |

Lexical Diversity) (McCarthy & Jarvis, 2010), which are more robust to varying text lengths compared to standard TTR (token-type ratio). Lower Maas scores and higher HD-D and MTLD scores indicate greater lexical diversity. We measure semantic diversity using inter- and intra-topic APS (average pairwise similarity) and INGF (inter-sample N-gram Frequency). For both APS and INGF, lower values signify higher diversity. For APS, we use cosine similarity to compute pairwise similarity.

**Comparison with existing datasets:** We compare ImPortPrompts to ROCStories (Mostafazadeh et al., 2016), WritingPrompts (Fan et al., 2018), and TinyStories (Li & Eldan). ImPortPrompts consists of 2000 samples[2] ($n = 2000$). To ensure consistent comparison, we randomly select an equal number of narratives from each other dataset to compute metrics. ImPortPrompts exhibits similar HD-D and Maas scores but significantly higher MTLD scores compared to others (Table 1). MTLD, being more sensitive to lexical diversity distribution throughout a text, suggests our narratives maintain more consistent diversity across their length. In terms of semantic diversity, ImPortPrompts shows comparable intra-topic APS and INGF scores, indicating similar within-topic and n-gram diversity. However, it demonstrates a notably higher inter-topic APS compared to ROCStories and WritingPrompts, suggesting more semantic similarity between narratives across topics.

Figure 3 compares the datasets in terms of character role representation and narrative length. The left plot shows that existing datasets significantly under-represent antagonist and victim roles compared to protagonists, which ImPortPrompts addresses with a more balanced distribution. The right plot shows narrative length distributions, where most of the stories in our dataset are concentrated between 5-30 sentences, making it suitable for analyses of short to medium-length narratives. ImPortPrompts exhibits a wider spread than TinyStories and the uniform 5-sentence structure of ROCStories (not depicted in the Figure), while avoiding the high variability of WritingPrompts.

In sum, ImPortPrompts offers advantages over existing datasets in terms of greater cross-topic similarity, greater lexical diversity across different text lengths, and improved representation of character roles and sentence lengths (particularly in short to medium-length texts). Furthermore, our dataset is specifically tailored to the task at hand by ensuring that characters are portrayed implicitly rather than explicitly. This enables us to better measure the ability to uncover implicit portrayal in long-tailed character roles, such as antagonists and victims, and to remove explicit portrayal confounders. In the next section, we use ImPortPrompts to assess LLMs' capability in uncovering implicit character portrayal.

---

[2]Each sample represents one narrative. The total number of *labels* is substantially higher, as each narrative contains up to five characters, each with a separate IAP label.

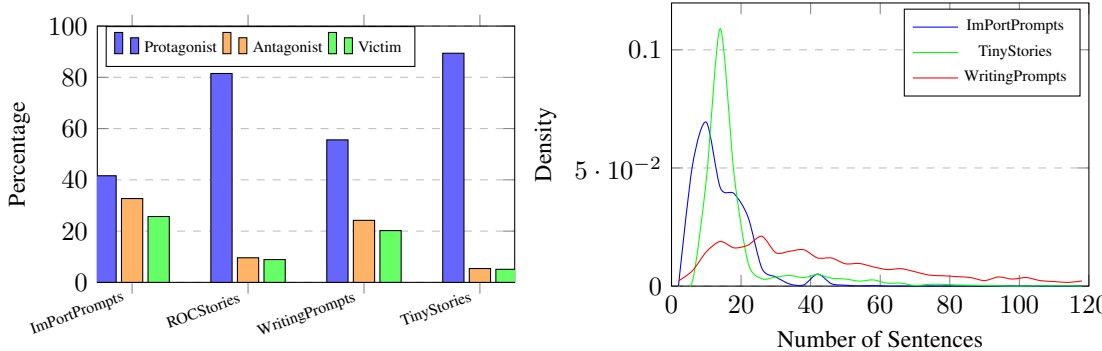

Figure 3: **Character Role Representation and Sentence Count Distribution Across Datasets:** Existing datasets show a strong bias towards protagonists, while our dataset contains a more balanced distribution. Our dataset covers a broader range of short-medium length texts compared to TinyStories and ROCStories (contains only 5-sentence narratives, not depicted in Figure), while avoiding the high variability in WritingPrompts.

## 3 UNCOVERING CHARACTER PORTRAYAL FROM NARRATIVE TEXTS

Given a narrative text $x^{(i)}$ with constraints $C_i$, our primary goal is to classify the intellect, appearance, and power of each of the $N_c$ characters into $\{\text{low}, \text{neutral}, \text{high}\}$. This can be defined by a function $f : \mathcal{X} \times \mathcal{C} \to \mathcal{Y}$ where $y_{c_j}^{(i)} = f(x^{(i)}, c_j)$. i.e. $y_{c_j}^{(i)}$ is the label set for character $c_j$ in narrative $i$.

### 3.1 LIIPA: LLMS FOR INFERRING IMPLICIT PORTRAYAL FOR CHARACTER ANALYSIS

We start by comparing the efficacy of LLMs against the method of Huang et al. (2021), which used COMET (Bosselut et al., 2019) as an auxiliary model to infer implicit character portrayal. We refer to this baseline as **COMET-Implicit Character Portrayal (COMET-ICP)**. We hypothesize that LLMs offer superior performance in this task because of their comprehensive world knowledge acquired through extensive pretraining and their capacity to effectively leverage long-range context for making predictions. COMET is limited to processing sentences with a simple event structure and generating a set of attributes describing the subject of the sentence. For instance, given the sentence: *"Alice gave Bob a cup of coffee."*, it may output the character attribtue word list $\hat{z}_w = \{$*generous, kind, thoughtful*$\}$.

**Evaluation Methodology and Experimental Setup:** We compare three LLM-based approaches against COMET-ICP (Figure 4). With LIIPA-STORY and LIIPA-SENTENCE, we prompt the LLM to generate character attribute word lists ($\hat{z}_w$) from complete stories and individual sentences respectively, mirroring COMET's output format. With LIIPA-DIRECT, we prompt the LLM to directly classify character portrayal based on the entire narrative, bypassing the wordlist generation. We refer to this entire framework as LIIPA.

Our choice of classification LLM is Google's Gemini (Georgiev et al., 2024). We choose a different LLM family from those used to generate narratives (GPT/Claude) to avoid self-preference bias, a phenomenon where LLM evaluators recognize and favor their own generations (Panickssery et al.). All prompts for this section can be found in §B.1. To ensure reproducible results, we set the LLM temperature parameter to 0 for all experiments in this section.

To assess the quality of generated character attribute wordlists for uncovering a character's IAP, we use LLMs-as-a-judge (Zheng et al., 2023) which involves prompting a separate evaluation LLM to infer a character's IAP solely from the generated wordlist. The underlying premise is that a high-quality wordlist should contain enough relevant information to enable accurate IAP inference. By applying LLMs-as-a-judge to

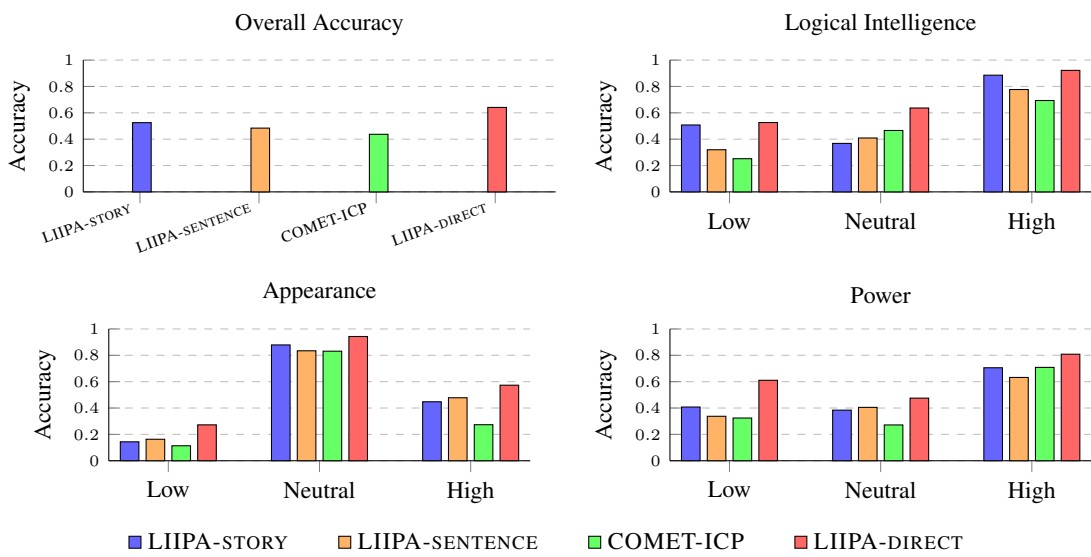

Figure 4: **Accuracy Across Portrayal Dimensions:** Clear hierarchy in model performance: LIIPA-DIRECT > LIIPA-STORY > LIIPA-SENTENCE > COMET-ICP. This trend highlights the importance of utilizing full narrative context. Directly prompting an LLM to uncover character portrayal (LIIPA-DIRECT) yields optimal results, as opposed to mapping character attribute word lists to labels. See Section 3.1 for definitions. ($n = 2000$)

wordlists generated by LLMs and COMET-ICP, we can quantitatively compare their effectiveness. If LLMs-as-a-judge achieves higher performance on LLM-generated wordlists compared to COMET-ICP-generated ones, we can conclude that the LLM wordlists are superior indicators of a character's IAP. Our choice of LLM-judge is GPT-4. Note that although we used the GPT LLM family for narrative generation, we still avoid self-preference bias since this model is mapping Gemini-generated wordlists to labels without access to the underlying GPT-generated narratives. For a discussion on self-preference bias, see §C.1.

**Accuracy Comparison**: Figure 4 presents our accuracy results across all three portrayal dimensions. LIIPA-DIRECT consistently outperforms the other methods across all dimensions and labels, indicating that directly prompting the model to generate character portrayal labels is more effective than the predominant approach of mapping character attribute wordlists to labels.

We observe that LIIPA-SENTENCE is more accurate than COMET-ICP, suggesting that LLMs can generate more informative character attribute wordlists when used as a direct substitute for COMET. Furthermore, LIIPA-STORY is more accurate than LIIPA-SENTENCE, indicating that the additional context gained from using the entire narrative as input (rather than individual sentences) helps in generating more informative wordlists. This may be because when analyzing a full story, the contextual methods can identify character arcs, interactions between characters, and how traits are revealed over time, whereas sentence-level analysis might miss these broader narrative elements. We also observe that accuracy varies significantly depending on the dimension and label. For instance, the appearance dimension shows the most varied performance across labels, with all methods performing much higher when classifying neutral compared to low and high. This may indicate that the model tends to refrain from making definitive classification decisions regarding appearance, in contrast to intelligence and power. This could also indicate that appearance descriptions are more subjective or culturally dependent, making them more challenging for models to classify consistently. Additionally, we note that the methods generally perform better on high versus low character portrayal suggesting that the methods are better at detecting implicit cues of positive character portrayal.

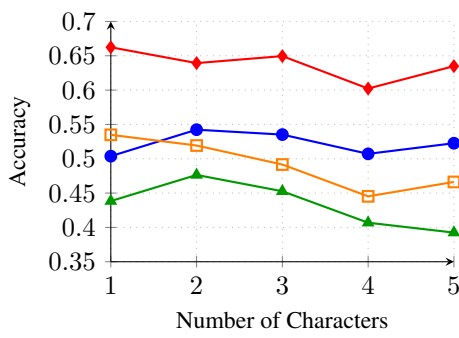 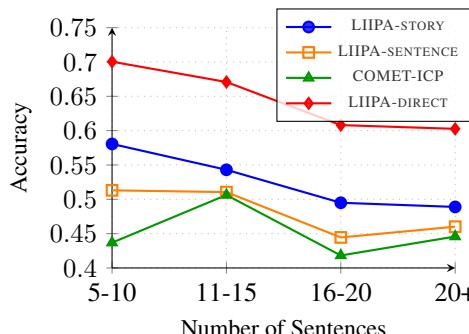

Figure 5: **Impact of Narrative Complexity on Accuracy.** Non-contextual methods (COMET-ICP, LIIPA-SENTENCE) show a stronger negative correlation with an increasing number of characters, while contextual methods demonstrate greater robustness. Conversely, contextual methods exhibit a negative correlation with an increasing number of sentences. Contextual methods outperform non-contextual methods in nearly all cases, with LIIPA-DIRECT consistently maintaining a significant performance advantage across varying narrative complexities. ($n \approx 300, 100$ per point for left/right plots respectively.)

In sum, our findings reveal a clear trend in model accuracy (LIIPA-DIRECT > LIIPA-STORY > LIIPA-SENTENCE > COMET-ICP). They underscore the importance of full narrative context, highlight the challenges in classifying appearance compared to other attributes, and demonstrate a general bias towards accurately detecting positive character portrayals across all methods.

**Impact of Narrative Complexity on Accuracy**: Figure 5 illustrates how increasing narrative length and number of characters affects model performance. As the number of characters increases, we observe that the non-contextual methods (COMET-ICP and LIIPA-SENTENCE) exhibit a stronger negative correlation compared to the contextual methods (LIIPA-STORY and LIIPA-DIRECT). The latter are also more robust to increases in character count, suggesting that having full narrative information is beneficial for stabilizing performance across an increasing number of characters. In contrast, we see that the contextual methods have a stronger negative correlation with the number of sentences, indicating that *longer narratives can actually hurt performance*. This might be due to an "information overload," where the model has to process and integrate a larger amount of information when making its classification decision. The non-contextual models don't suffer as much from this issue, perhaps because of their sentence-level processing.

These plots further illustrate the wide performance gap between the LIIPA-DIRECT method and the other methods, demonstrating the utility of bypassing wordlist generation and simply prompting an LLM to uncover character portrayal. In all cases (except for 1-character narratives), the contextual methods outperform the non-contextual methods, highlighting the importance of having full narrative context for uncovering character portrayal.

While LIIPA-DIRECT outperforms the other methods in terms of accuracy, this may come with a fairness cost which we investigate in the next section.

## 3.2 FAIRNESS IMPLICATIONS OF LLM-BASED CHARACTER PORTRAYAL INFERENCE

**Metrics and Experimental Setup:** We aim to ensure consistent performance of LLMs in character portrayal analysis across diverse demographic backgrounds, such as a black female antagonist or a disabled protagonist. We investigate this by prompting an LLM to insert character socio-demographic information (from Table 18) into our anonymized dataset (Tamkin et al.). We then measure disparate model performance when using LIIPA to classify implicit character portrayal. To get an overall estimate for disparity across de-

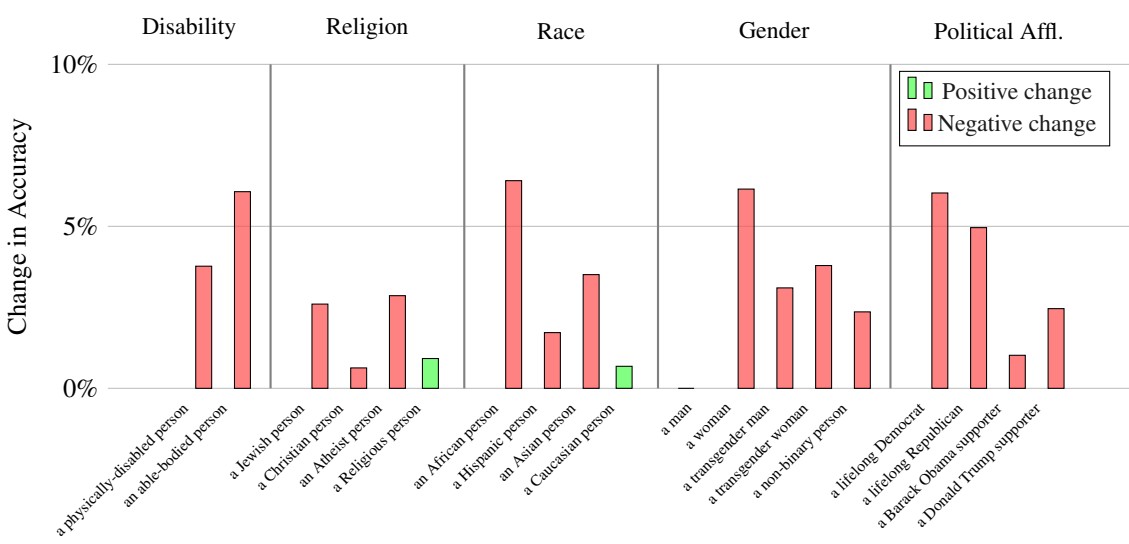

Figure 6: **Accuracy Disparities across Demographic Groups:** Change in accuracy after inserting demographic information when using LIIPA-DIRECT for portrayal label generation. Red bars indicate decreased accuracy, green bars show increased accuracy. Notable variance in accuracy changes observed within and across demographic categories. In nearly all cases, inserting demographic information results in an accuracy drop. ($n \approx 300$ per demographic category)

mographic groups (e.g., gender), we compute the variance in accuracy between group members (e.g., man, woman) and then average these variances across groups.

**Accuracy Disparities across Demographic Groups:** Figure 6 reveals significant accuracy disparities between group members when demographic information is inserted. We focus on the power portrayal dimension but find similar disparity patterns across appearance and intelligence (§B.2). For instance, assigning a character as a woman results in a ∼6% accuracy drop, while no such drop occurs for men. Rare instances of *"positive discrimination"* emerge: characters identified as religious or Caucasian show slight accuracy increases, contrasting with significant drops for the other group members. Across all three portrayal dimensions, incorporating demographic information significantly reduces model performance. The most substantial decline occurs when "low" portrayals are misclassified as "neutral" (344 instances, 3.14%), highlighting that the model tends to favor neutral predictions when demographic attributes are present. We now examine how these demographic disparities manifest across the various portrayal detection methods previously analyzed.

**Fairness-Accuracy Tradeoff:** Figure 7 illustrates a fairness-accuracy tradeoff for the various methods used in our LIIPA framework. The squares represent different prompting strategies used in LIIPA-DIRECT, each requiring different amounts of intermediate computation (e.g., chain of thought, tree of thought, etc.). LtM represents least-to-most prompting (Zhou et al., 2023) while DP represents prompting to directly generate labels without any intermediate computation. LIIPA-DIRECT yields higher accuracy but lower fairness, while word list approaches (LIIPA-SENTENCE, LIIPA-STORY, COMET-ICP) offer increased fairness at the cost of accuracy. Notably, our LLM-based word list approaches outperform the COMET-ICP in both accuracy and fairness. Thus, the curve illustrates a trade-off between fairness and accuracy, reflecting how increased contextual information and intermediate computation tend to increase performance but potentially at a cost to fairness.

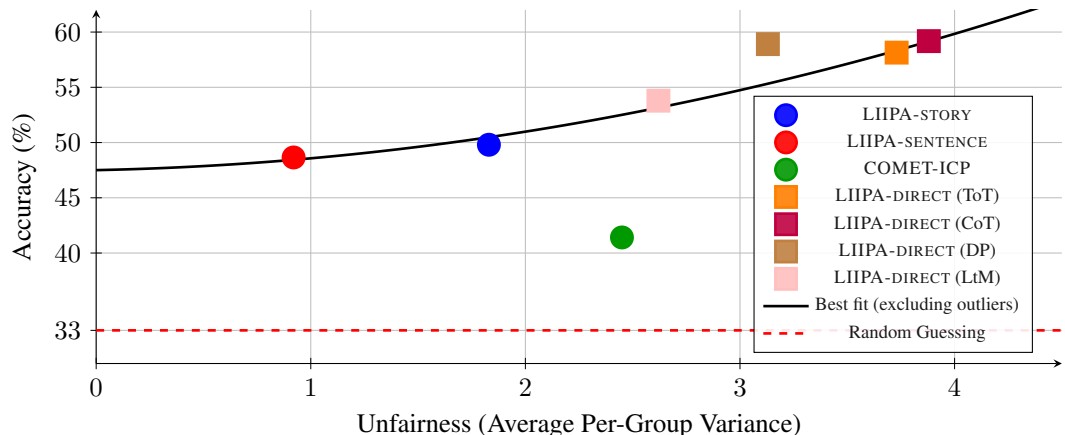

Figure 7: **Fairness-Accuracy Tradeoff**. Circles denote word list-based methods (with and without use of an LLM), squares indicate directly prompting an LLM to uncover portrayal (LIIPA-DIRECT) under various prompting strategies. We identify a fairness-accuracy tradeoff for LLM approaches where LIIPA-DIRECT achieves higher accuracy, while LLM word list approaches (LIIPA-STORY, LIIPA-SENTENCE) achieve lower unfairness. ($n = 2000$)

## 4 RELATED WORKS

**Character Portrayal Tools:** Visualization tools for character portrayal have demonstrated significant benefits for both writers and literary scholars. Hoque et al. (2023) developed *Portrayal*, a character visualization tool, and conducted a user study revealing its effectiveness in helping writers revise drafts and create more dynamic characters. The tool also aided scholars in developing tangible evidence to support literary arguments. In a separate study, Hoque et al. (2022) interviewed writers and found they struggle to track implicit character biases, especially in longer, more complex narratives. This insight led to the development of *DramatVis Personae*, a tool designed to help writers more easily identify various biases, including those related to character portrayal. Our LIIPA framework can enhance these tools by providing writers with deeper insights into the implicit portrayal of their characters.

**Measuring Fairness:** Our approach to fairness measurement focuses on disparate accuracy across demographic groups, a method previously used in studies examining implicit biases in LLMs (Gupta et al., 2024), and character portrayal in AI-generated texts (Huang et al., 2021). To facilitate this analysis, we used LLMs to insert demographic information into narratives. This technique aligns with recent work: Tamkin et al. used LLMs to add demographic attributes to LLM-generated data for evaluating discrimination, while Perez et al. (2023) used LLMs to generate evaluation data for uncovering harmful model behaviors.

## 5 CONCLUSION

We have proposed a new framework called LIIPA that uses LLMs to infer implicit character portrayal within narrative text. LIIPA outperforms non-LLM character portrayal estimation in both accuracy and fairness while being robust to longer texts with more characters. However, the identified fairness-accuracy tradeoff underscores the need for cautious application of LLMs when estimating character portrayal. We also introduced ImPortPrompts, a dataset for character portrayal estimation that offers improved diversity and greater cross-topic similarity over existing benchmarks. Future work can apply LIIPA to better understand how implicit portrayal biases manifest in narratives and to improve portrayal visualization tools such as those developed by Hoque et al. (2023).

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

# Appendix

## Table of Contents

# A DATASET GENERATION DETAILS

## A.1 NARRATIVE DEFINITIONS

| Dimension | Definition |
|---|---|
| Logical Intelligence | The ability to think conceptually and abstractly, and the capacity to discern logical and numerical patterns. |
| Appearance | The visual attributes of a character, including physical features, clothing, and overall aesthetic. |
| Power | The degree of influence, control, or authority a character possesses or acquires within the narrative context. |

**General Classification Information:**
For each dimension, a character's portrayal should be classified as low, neutral, or high based on the information provided in the narrative and their development arc:

- **Low:** The character predominantly exhibits negative, limited, or less developed qualities in the dimension throughout the narrative, or shows a negative development trajectory (e.g., from high to low).

- **Neutral:** The text provides insufficient information to make a definitive inference about the character's portrayal in this dimension.

- **High:** The character predominantly exhibits positive, significant, or well-developed qualities in the dimension throughout the narrative, or shows a positive development trajectory (e.g., from low to high).

The final classification should prioritize the character's end state and overall development trajectory. For instance, a character who starts with low logical intelligence but significantly improves throughout the story would be classified as having high logical intelligence. Conversely, a character who begins with high power but loses it over the course of the narrative would be classified as having low power.

Table 2: Definitions of Character Portrayal Dimensions and Classification Guidelines

## A.2 CHARACTER ROLE SAMPLING ALGORITHM

**Intuition:** For a single character (n=1), it's always assigned as the Protagonist. This makes sense as a story typically needs a main character. For two characters (n=2), the algorithm assigns a Protagonist and an Antagonist. This creates a basic conflict structure common in many narratives. For three characters (n=3), it assigns one of each role: Protagonist, Antagonist, and Victim. This allows for a more complex narrative structure with clear roles. For more than three characters (n > 3), the algorithm ensures at least one of each role is present, then randomly assigns additional roles. This maintains narrative balance while allowing for variation and complexity in larger casts.

## A.3 FULL CONSTRAINT LIST

1. The narrative contains exactly $N_c^{(i)}$ characters.

2. The narrative length is $L^{(i)}$ sentences.

3. Each character $j$ is assigned a role from {protagonist, antagonist, victim} following Algorithm 1.

4. Each character $j$ is assigned a portrayal label set $y_j^{(i)} \in \mathcal{Y}$.

| Role | Definition |
|------|------------|
| Protagonist | A main character in the story who plays a central role in driving the plot forward. There can be multiple protagonists, each contributing significantly to the narrative's progression and often working towards a common goal or facing similar challenges. |
| Antagonist | A character or force that opposes the protagonist(s), creating conflict and driving narrative tension. Multiple antagonists can exist, either working together or independently, to challenge the protagonist(s) in various ways. |
| Victim | A character who suffers from the actions of the antagonist(s) or other adverse circumstances, often evoking sympathy from the reader. There can be one or more victims in a story. |

Table 3: Definitions of Character Roles

---

**Algorithm 1** Character Role Sampling

---

**Require:** $n > 0$ (number of characters)
**Ensure:** A set $S$ of $n$ character roles
1: $R \leftarrow \{$Protagonist, Antagonist, Victim$\}$
2: **if** $n = 1$ **then**
3:     $S \leftarrow \{$Protagonist$\}$
4: **else if** $n = 2$ **then**
5:     $S \leftarrow \{$Protagonist, Antagonist$\}$
6: **else if** $n = 3$ **then**
7:     $S \leftarrow R$
8: **else**
9:     $S \leftarrow R$                                                                    ▷ Ensure at least one of each role
10:     **while** $|S| < n$ **do**
11:         $role \leftarrow$ Random sample from $R$
12:         $S \leftarrow S \cup \{role\}$
13:     **end while**
14: **end if**
15: **return** $S$

---

5. **Implicit portrayal:** The portrayal of each character must be revealed implicitly through their actions, decisions, and interactions, rather than through explicit words and statements. For each of the three portrayal categories, the narrative should avoid using the following words directly to describe characters:

   - **Intellect**: brilliant, intelligent, smart, clever, wise, intellectual, genius, knowledgeable, analytical, logical
   - **Appearance**: beautiful, handsome, attractive, ugly, pretty, gorgeous, plain, stunning, hideous, charming

- **Power**: powerful, influential, dominant, weak, strong, authoritative, powerless, commanding, subordinate, forceful

6. The socio-demographic background of characters should not be explicitly stated or implied. Specifically:

    - **Character Naming:** Refer to characters as [Role]X, where Role is Protagonist, Antagonist, or Victim, and X is a unique identifier (e.g., Protagonist1, Antagonist2).
    - **Gender:** Use gender-neutral language throughout. Avoid gendered pronouns (he/she) and titles (Mr./Mrs./Ms.). Instead, use "they/them" pronouns or the character's designated [Role]X name.
    - **Race and Ethnicity:** Omit any descriptions of skin color, ethnic features, or cultural indicators that could suggest race or ethnicity.
    - **Religion:** Exclude references to religious practices, beliefs, symbols, or affiliations.
    - **Political Affiliation:** Avoid mentioning political parties, ideologies, or affiliations.
    - **Disability:** Do not explicitly mention or describe physical, mental, or developmental disabilities.

7. The narrative genre and topic are selected from Table 4.

## A.4 NARRATIVE GENRES AND TITLES

| Genre | Titles |
|---|---|
| Fantasy | The Enchanted Forest, Dragon's Quest, The Sorcerer's Stone, Tales of Avalon, The Elven Kingdom |
| Science Fiction | Journey to Mars, The AI Revolution, Galactic Wars, The Time Machine, Alien Encounters |
| Mystery | The Secret Detective, The Vanishing Act, Murder at the Mansion, The Hidden Clue, The Enigma Code |
| Thriller | The Chase, Undercover Agent, The Last Witness, The Hostage Situation, The Dark Conspiracy |
| Romance | Love in Paris, The Heart's Desire, The Secret Admirer, A Summer Romance, The Wedding Planner |
| Historical Fiction | The Roman Empire, A Tale of Two Cities, The Civil War Diaries, The Renaissance Man, The Samurai's Honor |
| Horror | The Haunted House, The Vampire's Curse, The Ghost in the Attic, The Witching Hour, The Monster in the Closet |
| Adventure | The Lost Treasure, Expedition to the Amazon, The Pirate's Cove, The Mountain Climb, The Jungle Survival |
| Drama | The Family Secret, The Broken Dream, The Great Betrayal, The Healing Journey, The Final Performance |
| Comedy | The Misadventures of Tom, The Office Prank, The Wedding Fiasco, The Awkward Date, The Clumsy Hero |

Table 4: Narrative Genres and Titles

## A.5 TREE-OF-THOUGHTS PROMPTING FOR STORY GENERATION

The styling of our prompts in the Appendix is inspired by Perez et al. (2023).

**System:** You are a skilled story planner. Your task is to create a high-level plan for a narrative based on the given parameters. Each character's portrayal can be defined and classified as follows:

- *[Insert character portrayal definitions from Table A.1]*

The character roles are defined as follows:

- *[Insert character role definitions from Table 3]*

*[Insert implicit portrayal constraint from Appendix A.3]*
*[Insert socio-demographic background constraint from Appendix A.3]*

Create a story plan for a *[GENRE]* genre story titled "*[TITLE]*". The story should have *[NUMBER]* characters: *[CHARACTER_ROLES]*. The narrative should be *[LENGTH]* sentences long.
Ensure that:

- *[CHARACTER1]* is portrayed with:
    - *[LEVEL]* logical intelligence
    - *[LEVEL]* appearance
    - *[LEVEL]* power
- *[CHARACTER2]* is portrayed with:
    - *[LEVEL]* logical intelligence
    - *[LEVEL]* appearance
    - *[LEVEL]* power
- *[repeat for each character]* ...

Remember that the "neutral" label means the text provides insufficient information to make a definitive inference about the character's portrayal.
Provide a high-level plan for generating the story that will satisfy all the provided constraints.

**Assistant:**

Table 5: Story plan generation prompt for ToT

**System:** You are an expert story analyst. Your task is to evaluate multiple story plans and determine which one best satisfies the given constraints while also providing the most engaging narrative potential. *[list definitions and constraints here as done in the previous Table]*

**Human:** Here is a list of story plans for a *[GENRE]* genre story titled "*[TITLE]*". The story should have *[NUMBER]* characters: *[CHARACTER_ROLES]*. The narrative should be *[LENGTH]* sentences long. The character portrayals should be:

- *[List character portrayals as done in the previous prompt]*

*[STORY_PLANS]*
Which plan best satisfies the constraints and offers the most engaging narrative potential? Explain your choice. Then, structure your final answer as: "Chosen Plan: Plan*[NUMBER]*"

**Assistant:**

Table 6: Story plan voting prompt for ToT

**System:** You are a skilled storyteller. Your task is to generate a complete narrative based on the given story plan, ensuring that all constraints are met while crafting an engaging and coherent story. *[list definitions and constraints here as done in the previous prompts]*

**Human:** Generate a *[GENRE]* genre story titled "*[TITLE]*" based on the following plan:
*[Insert the winning story plan here]*
*[Insert story generation constraints as done in previous prompts]*
Generate a complete narrative that follows this plan and meets all constraints.

**Assistant:**

Table 7: Narrative generation prompt for ToT

**System:** You are an expert story analyst. Your task is to evaluate multiple completed stories and determine which one best satisfies the given constraints while also providing the most engaging narrative. *[list definitions and constraints here as done in the previous Tables]*

**Human:** Here is a list of completed stories for a *[GENRE]* story titled *[TITLE]*.
*[List story constraints and character portrayals]*
*[List actual stories]*
Which story best satisfies the constraints and offers the most engaging narrative? Explain your choice. Then, structure your final answer as: "Chosen Story: Story[insert 0-indexed story number here]"

**Assistant:**

Table 8: Story voting prompt for ToT

## A.6 DATA VALIDATION PROCEDURES

We perform quality assurance on our dataset to ensure the LLM-generated narratives comply with the constraints outlined in §A.3. We address both lexical and semantic constraints through automated and human validation processes, respectively. Narratives failing either check are excluded from the final dataset.

**Automated Validation:** We programmatically verify character count and narrative length constraints using scripts available in our code repository. Characters follow a fixed structure (ProtagonistX/AntagonistX/VictimX, X being a number), which facilitates easy extraction. We partially perform semantic validation by using exclusion word lists to ensure narratives do not contain explicit portrayal indicators and demographic information.

**Human Validation:** We perform manual validation to ensure characters align with their assigned roles and are implicitly portrayed as per ground truth labels. We verify genre and title constraints, and confirm the absence of explicit or implicit demographic information. We manually validate a random subset of 100 narratives using instructions from §A.6.1, consistent with prior works (Dammu et al., 2024; Dahl et al., 2024) validating LLM-generated data.

Qualitative analysis reveals strong adherence to assigned genres, character roles, and portrayal constraints, with minimal demographic information leakage. While most narratives successfully avoid explicit portrayal indicators, some instances present borderline cases between implicit and explicit indicators (e.g., "their name carrying weight" or "charismatic wedding planner" for high power). However, overt explicit indicators remain rare.

### A.6.1 ANNOTATION TEMPLATE

```
Narrative ID: [Insert unique identifier for the narrative]

1. Character Role Verification:
   For each character (protagonist0, antagonist0, victim0, etc.):
     Assigned role: [Protagonist/Antagonist/Victim]
     Role fulfilled in narrative: [Yes/No]
     If No, explain discrepancy: [Explanation]

2. Character Portrayal Consistency:
   For each character:
     Intellect portrayal: [Low/Neutral/High]
     Appearance portrayal: [Low/Neutral/High]
     Power portrayal: [Low/Neutral/High]

3. Absence of Socio-demographic Information:
   For each character:
       Socio-demographic info present: [Yes/No]
       If Yes, describe: [Explanation]

4. Genre and Topic Adherence:
   Specified genre: [Genre]
   Specified topic: [Topic]
   Adheres to genre: [Yes/No]
   Adheres to topic: [Yes/No]
   If No to either, explain: [Explanation]
```

```
5. Overall Semantic Constraint Adherence:
   All semantic constraints met: [Yes/No]

6. Additional Comments:
    [Free text area for any other observations or notes]

Annotator ID: [Unique identifier for the annotator]
```

**Instructions for Annotators:**

- Fill out all fields in the interface for each narrative you review.
- For character role verification, assess whether each character's actions and interactions in the narrative align with their assigned role.
- In the "Additional Comments" section, note any unusual or interesting aspects of the narrative that aren't captured by the other fields.

### A.7 ADDITIONAL EXPERIMENTAL DETAILS

When generating narratives, we vary the number of characters, $N_c^{(i)}$ to be between 1 to 5 and the number of sentences, $L^{(i)}$ to be 5, 10, 15, or 20.

## B UNCOVERING CHARACTER PORTRAYAL DETAILS

### B.1 LIIPA PROMPTING

**System:** You are an AI assistant trained to generate a list of 5 character attributes that describe a specific character's personality, traits, or qualities based on the sentence provided. Format your answer like: [attr1, attr2, ...]

**Human:** Given the following **sentence**, generate a list of 5 attributes that describe *[CHARACTER]*'s personality or qualities. Provide your answer as a comma-separated list of attributes, focusing on *[CHARACTER]*'s portrayal throughout the sentence. Format your answer like: [attr1, attr2, ...]
Sentence: *[Insert SENTENCE here]*

   **Assistant:**

Table 9: LIIPA-SENTENCE Prompt

**System:** You are an AI assistant trained to generate lists of 5 character attributes that describe the personalities, traits, or qualities of all characters in a story based on the entire context provided. You will format your answer as a JSON object where each character is a key and their attributes are an array of strings.

**Human:** Given the following narrative, generate a list of 5 attributes for each character that describe their personality or qualities. Provide your answer as a JSON object where each character is a key and their attributes are an array of 5 strings. Focus on each character's portrayal throughout the entire narrative. Format your answer like this: *[formatting instructions]*
Output your answer and nothing else.
Narrative: *[Insert NARRATIVE here]*

    **Assistant:**

Table 10: LIIPA-STORY Prompt

**System:** You are an AI assistant trained to analyze character portrayals based on given lists of attributes. Your task is to infer each character's intellect, appearance, and power (IAP) solely from the provided attribute wordlists. Classify each aspect as either low, neutral, or high for each character. Each character's portrayal can be defined and classified as follows:

    • *[Insert character portrayal definitions from Table A.1]*

The character roles are defined as follows:

    • *[Insert character role definitions from Table 3]*

*[Insert formatting instructions]*

**Human:** Wordlist: *[Insert WORDLIST here]*

    **Assistant:**

Table 11: LLM-as-a-judge Prompt

**System:** You are an AI assistant trained to analyze character portrayals in narratives. Your task is to classify a character's intellect, appearance, and power (IAP) as low, neutral, or high based on the given narrative. Each character's portrayal can be defined and classified as follows:

    • *[Insert character portrayal definitions from Table A.1]*

**Human:** Given the following narrative, classify the intellect, appearance, and power (IAP) of each character as low, neutral, or high.
*[Insert formatting instructions]*
Narrative: *[Insert NARRATIVE]*

    **Assistant:**

Table 12: LIIPA-DIRECT Direct Prompting (DP)

**System:** You are an AI assistant trained to analyze character portrayals in narratives. Your task is to classify each character's intellect, appearance, and power (IAP) as low, neutral, or high based on the given narrative. Each character's portrayal can be defined as follows:

- *[Insert character portrayal definitions from Table A.1]*

For each character, provide a step-by-step reasoning process for your classification of their intellect, appearance, and power. After your reasoning, provide the final classifications as a JSON object where each character is a key and their IAP classifications are an array of three strings.

**Human:** Given the following narrative, analyze and classify the intellect, appearance, and power (IAP) of each character as low, neutral, or high. For each character, provide your step-by-step reasoning for each classification. Then, summarize your classifications in a JSON object where each character is a key and their IAP classifications are an array of 3 strings.
Narrative: *[Insert NARRATIVE]*

**Assistant:**

Table 13: LIIPA-DIRECT Chain of thought (CoT)

**System:** You are an AI assistant trained in task decomposition for concise narrative analysis. Your role is to break down complex character analysis tasks into sequential subproblems, focusing on Protagonists, Antagonists, and Victim character roles while emphasizing brevity and efficiency in the analysis process.

**Human:** Your task is to decompose the problem of classifying character portrayals (intellect, appearance, and power) from a given narrative into sequential subproblems, focusing specifically on Protagonists, Antagonists, and Victim roles. The final subproblem should be the actual classification for characters in these roles. Ensure that each subproblem builds on the previous ones, contributes to the final classification task, and emphasizes concise analysis and explanation.
Each character's portrayal can be defined and classified as follows:

- *[Insert character portrayal definitions from Table A.1]*

The character roles are defined as follows:

- *[Insert character role definitions from Table 3]*

Provide the decomposition as a numbered list of 3 subproblems, with the final one being the classification task. Each subproblem should emphasize concise analysis and explanation, avoiding unnecessary detail or repetition. Use the following format for each subproblem:
*[Subproblem formatting instructions]*

**Assistant:**

Table 14: LIIPA-DIRECT LtM Task Decomposition Prompt

**System:** You are an AI assistant trained to solve subproblems in sequential narrative analysis, focusing on Protagonists, Antagonists, and Victim character roles.

**Human:** Given the following narrative and the solutions to the previous subproblems, solve the current subproblem in the sequence for analyzing and classifying the portrayals of Protagonists, Antagonists, and Victims.
Narrative: *[NARRATIVE]*
Previous subproblem solutions: *[PREVIOUS SOLUTIONS]*
Current subproblem: *[SUBPROBLEM]*
Each character's portrayal can be defined and classified as follows:

- *[Insert character portrayal definitions from Table A.1]*

The character roles are defined as follows:

- *[Insert character role definitions from Table 3]*

Provide a detailed solution to the current subproblem, using the information from the narrative and the previous subproblem solutions. Ensure your solution directly contributes to the ultimate goal of classifying each character's intellect, appearance, and power as low, neutral, or high, with a focus on Protagonists, Antagonists, and Victims.

    **Assistant:**

Table 15: LIIPA-DIRECT LtM Subproblem Solving Prompt

**System:** You are an AI assistant trained to create classification plans for analyzing character portrayals in narratives. Your task is to generate a detailed plan for classifying characters' logical intelligence, appearance, and power (IAP) based on the given narrative.
Each character's portrayal can be defined and classified as follows:

- *[Insert character portrayal definitions from Table A.1]*

The character roles are defined as follows:

- *[Insert character role definitions from Table 3]*

**Human:** Generate a concise classification plan for analyzing the logical intelligence, appearance, and power (IAP) of all characters in the following narrative:
*[insert NARRATIVE]*
Your plan should briefly outline the steps you would take to classify each character's IAP as low, neutral, or high. Be specific but concise about what aspects of the narrative you would analyze and how you would use them to make your classifications.

    **Assistant:**

Table 16: LIIPA-DIRECT ToT Classification Plan Generation Prompt

1128
1129
1130
1131
1132
1133
1134
1135
1136
1137
1138
1139
1140
1141
1142
1143
1144
1145
1146
1147
1148
1149
1150
1151
1152
1153
1154
1155
1156
1157
1158
1159
1160
1161
1162
1163
1164
1165
1166
1167
1168
1169
1170
1171
1172
1173
1174

**System:** You are an AI assistant trained to execute classification plans for character portrayal analysis. Your task is to follow the given plan and classify the characters' logical intelligence, appearance, and power (IAP) as low, neutral, or high.
Each character's portrayal can be defined and classified as follows:

   • *[Insert character portrayal definitions from Table A.1]*

The character roles are defined as follows:

   • *[Insert character role definitions from Table 3]*

**Human:** Execute the following classification plan for analyzing the logical intelligence, appearance, and power (IAP) of all characters in the given narrative:
Narrative: *[insert NARRATIVE]*
Classification Plan: *[insert PLAN]*
Follow the plan step by step and provide your final classification for logical intelligence, appearance, and power as low, neutral, or high for each character.

   **Assistant:**

Table 17: LIIPA-DIRECT ToT Classification Plan Execution Prompt

## B.2 ADDITIONAL FAIRNESS PLOTS

Here, we show further measurements of accuracy stratified across different demographic groups, this time looking at the other two label types: intellect and appearance.

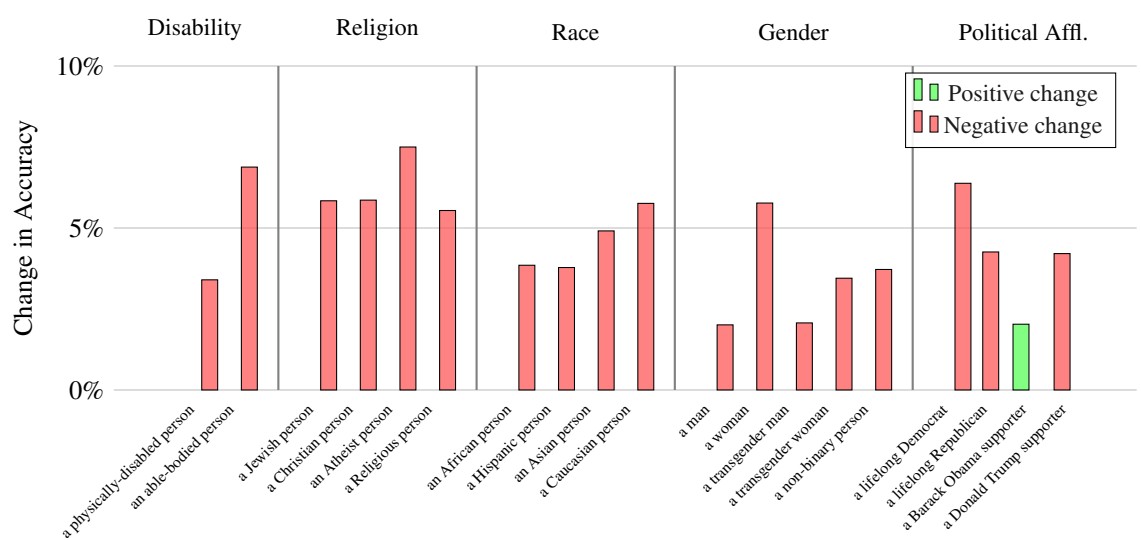

Figure 8: Accuracy Disparities across Demographic Groups for Intellect Dimension

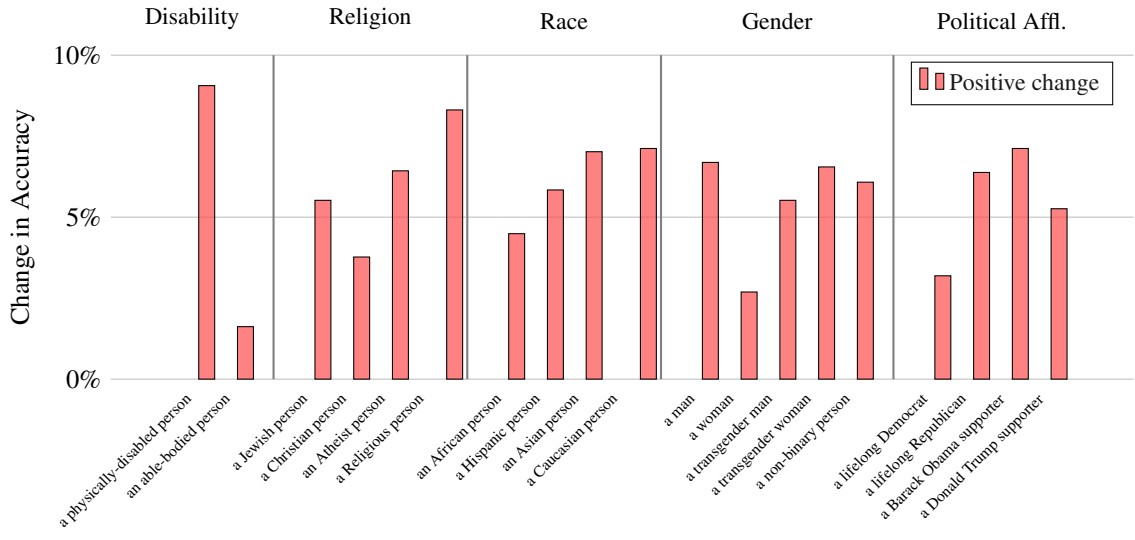

Figure 9: Accuracy Disparities across Demographic Groups for Appearance Dimension

## C MISC

### C.1 LLM SAMPLING PROCESSES AND AVOIDING SELF-PREFERENCE BIAS

Here, we recap the various processes in which we sample from LLMs in our work. Our LLM sampling processes include generating narratives from constraints (Sec 2.2), sampling character-attribute word lists, and sampling IAP labels either from word lists or directly from narratives and characters. This can be formally described below:

1. **Dataset Sampling:** $x^{(i)} \sim p_g(\cdot|C_i)$

2. **Word list Sampling:** $w_j^{(i)} \sim p_w(\cdot|x^{(i)}, c_j)$ For a given character $c_j$ in narrative $x^{(i)}$, we sample a word list $w_j^{(i)}$ from LLM $p_w$. This is the output format of LIIPA-SENTENCE and LIIPA-STORY.

3. **Label Sampling:** There are two separate sampling processes for IAP label generation:

   - **Word list-based Sampling (LLM Judge):** $y_j^{(i)} \sim p_l(\cdot|w_j^{(i)})$ For a given character $c_j$ in narrative $x^{(i)}$, we sample IAP labels $y_j^{(i)}$ from LLM $p_l$, conditioned on the generated word list $w_j^{(i)}$. This is used to "judge" the word lists generated by LIIPA-SENTENCE and LIIPA-STORY.
   - **Narrative-based Sampling:** $y_j^{(i)} \sim p_m(\cdot|x^{(i)}, c_j)$ We also sample IAP labels $y_j^{(i)}$ from LLM $p_m$, conditioned on the full narrative text $x^{(i)}$ and a given character $c_j$. This is the output format of LIIPA-DIRECT.

To avoid self-preference bias, a phenomenon where LLM evaluators recognize and favor their own generations, we must ensure the LLM model family responsible for label generation ($p_l$ and $p_m$) are distinct from the families used for narrative ($p_g$) and word list generation ($p_w$). We initialize $p_g$ to be GPT, Claude, and $p_w$ to be GPT. For label generation, we initialize $p_l$ and $p_m$ to be Google Gemini. Thus, we ensure that the label-generating LLM evaluates the content objectively, without favoring its own prior outputs which helps maintain the integrity of our evaluations and supports the validity of our findings.

### C.2 FAIRNESS MEASUREMENT DETAILS

| Group | Personas |
|---|---|
| Disability | a physically-disabled person, an able-bodied person |
| Religion | a Jewish person, a Christian person, an Atheist person, a Religious person |
| Race | an African person, a Hispanic person, an Asian person, a Caucasian person |
| Gender | a man, a woman, a transgender man, a transgender woman, a non-binary person |
| Political Affl. | a lifelong Democrat, a lifelong Republican, a Barack Obama Supporter, a Donald Trump Supporter |

Table 18: The 19 Personas across 5 socio-demographic groups that we explore in this study. Underlined words denote short forms used in tables for brevity, e.g., Phys. Disabled, Trump Supp., etc. Copied verbatim from Gupta et al. (2024).

