# OpenReview forum: "Show, Don't Tell: Uncovering Implicit Character Portrayal using LLMs"
_ICLR.cc/2025/Conference — ICLR 2025 Conference Withdrawn Submission_

### Official Review · Reviewer_8hkT · 2024-10-22

**Soundness:** 3
**Presentation:** 3
**Contribution:** 2
**Rating:** 5
**Confidence:** 3

**Summary:**

The paper presents a novel framework, LIIPA (LLMs for Inferring Implicit Portrayal for Character Analysis), that uses LLMs to uncover implicit character portrayals in narrative texts.

Contributions:
ImPortPrompts, a dataset for implicit character portrayal analysis.
LIIPA, a framework for prompting LLMs to uncover character portrayals.
Through analysis and investigation of performance and fairness using the framework.

This work addresses the gap in existing tools, which primarily rely on explicit textual indicators for character analysis. By generating a diverse dataset and implementing various LLM prompting techniques, the authors show that LIIPA outperforms existing methods in robustness and accuracy, particularly in complex narratives with multiple characters. Additionally, the paper explores the fairness-accuracy tradeoff in portrayal predictions across different demographic groups.

**Strengths:**

The paper introduces LIIPA, a novel framework designed to enhance the analysis of implicit character portrayal in narrative texts by leveraging LLMs. The framework, which outperforms prior methods like COMET-ICP, features multiple approaches: LIIPA-DIRECT, LIIPA-STORY, and LIIPA-SENTENCE, each employing different levels of intermediate computation for classification based on traits like intellect, appearance, and power.
The creation of ImPortPrompts, a dataset tailored for implicit portrayal analysis, highlights the paper’s effort to enable cross-topic diversity and improved representation of character roles. One of LIIPA’s key strengths lies in its ability to accurately capture portrayal across full narrative contexts, outperforming non-contextual approaches and proving resilient even as the complexity of character interactions increases. Additionally, the ablation study provides valuable insights into bias and fairness, acknowledging a fairness-accuracy tradeoff where accuracy gains might affect demographic fairness.

**Weaknesses:**

**Complexity of Portrayal Dimensions**: Focusing on three main portrayal dimensions (intellect, appearance, power) might oversimplify the complexity of character traits. The three papers authors cited in task formulation (Lucy & Bamman, 2021) and (Huang et al., 2021) were focused on gender bias, so it is understandable to represent gender-related characteristics under these dimensions. (Huang et al., 2024) included a dimension set of six since they focused on a more general analysis. As the authors focused on general character portrayal analysis, would three dimensions potentially overlook other significant attributes like emotional depth or moral alignment?

**Benchmark dataset**: As mentioned in the last paragraph of the introduction, authors tend to introduce ImPortPrompts as "the first benchmark dataset designed specifically for this task". Have the authors considered verifying the effectiveness of this benchmark on different sizes of LLMs and maybe verifying the alignment between human annotators and benchmark results based on a subset of outputs?

**Questions:**

**Fairness Evaluation**: The discussion in the Fairness-Accuracy Tradeoff section appears somewhat ambiguous. Based on Figure 7, higher accuracy methods correlate with increased unfairness. Does this suggest that the ground-truth labels themselves might carry implicit biases, leading to unfairness in model performance? Alternatively, is the observed unfairness solely a consequence of how the LLM processes and interprets narratives, independent of the fairness of the training data?

---

### Official Review · Reviewer_bQ8v · 2024-11-01

**Soundness:** 1
**Presentation:** 2
**Contribution:** 2
**Rating:** 3
**Confidence:** 4

**Summary:**

This paper presents LIIPA, a set of methods for measuring the implicit portrayals of characters along the dimensions of appearance, intellect, and power. As a test bed, authors create a dataset of 2000 stories containing characters portrayed with various levels of intellect/power/appearance. LIIPA methods prompt LLMs to either generate the portrayal classification directly, or  generate intermediate word lists describing the character, and then classifying the portrayal. Results show that the story level method outperforms the sentence-level methods. Additionally, authors conduct investigations into the effect on classification performance of adding demographic attributes into the stories, finding that most demographic identities see a decrease in performance.

**Strengths:**

- The main problem tackled by the paper, i.e., that of quantifying and analyzing implicit portrayals of characters, is very interesting.
- I liked the creation of a highly controlled dataset for controlled explorations.
- I liked the investigations around fairness that the authors conducted.

**Weaknesses:**

Despite liking the overall goals of the paper, I found that the paper had several strong weaknesses that prevent the paper to be publication-ready in my eyes.

1. While I appreciate the focus on implicit portrayal with the creation of a synthetic dataset, there is severe issue of external validity of the experiments.
   1. In real narratives, characters are both implicitly and explicitly portrayed; creating a dataset where only implicit portrayal present needs to be better motivated.
   2. Furthermore, the fact that experiments are only done on the synthetic data call into question the usefulness or validity of methods on real datasets. How would the LIIPA method perform on real world datasets? I wish authors had done (even a small-scale) investigation of this method.
2. Relatedly, part of the issue is that the synthetic data is only loosely validated.
   1. While authors mention a human validation in the appendix (this should really be mentioned in the main body of the paper), they do not state the results of that validation. How many stories did not adhere to the templates?
   2. L148: Machine generation is notoriously biased against certain demographic groups, as evidenced in the Lucy & Bamman paper authors keep citing. This directly contradicts the claim that synthetically generated stories have less representational biases.
   3. Furthermore, given the biases in performance for "negative" portrayals (e.g., low appearance, low intellect, etc.), it seems feasible that the stories generated for these negative portrayals aren't actually that negative?
3. The discussion of prior work is very limited (partly evidenced by the very small amount of papers cited in the related work section of the paper).
   1. There have been many works examining implicit portrayals of characters and biases in the portrayal of characters (https://www.aclweb.org/anthology/D17-1247, https://www.frontiersin.org/articles/10.3389/frai.2020.00055/full, https://aclanthology.org/P19-1243/, https://aclanthology.org/P18-1043/, https://dl.acm.org/doi/abs/10.1145/3359190, etc. ). While some operate at the lexical level, their analyses are aggregated at the character level over full stories or documents, making them useful at larger contexts.
   2. L136: Relatedly, this is a mischaracterization of prior work. While some prior work only examined protagonists, many of the references above examine many characters in stories.
4. I wish the authors expanded more on why they chose the three dimensions of intellect, appearance, and power.
   1. The paper glosses over why these dimensions are relevant to character portrayals, simply citing three previous works to back up that choice. But there is a wealth of humanities and social science literature that motivates why certain aspects of character portrayal dimensions matter / are salient that authors should explore and cite.
   2. Relatedly, what does high vs. low appearance mean here? Authors should provide more details.
5. There are issues around the dataset comparison:
   1. It feels unfair to compare the newly generated dataset to other existing resources since some of these are explicitly not meant to be diverse (e.g., ROC stories is meant to be short commonsense stories only, TinyStories is machine-generated stories for children), and artificially inflates the qualities of the dataset... I wish authors did a literature survey of open-source narrative/story datasets to make fairer comparisons (e.g., TorontoBookCorpus, FanFic corpus, and others).
   2. L221: Authors argue that their dataset has a better length distribution than other datasets they compared to, but I don't see why that is true? There is no reason to believe 5-30 sentences is the "right length" for stories, and in fact, I'd argue many stories are much longer (e.g., books).
   3. L228: I don't agree that antagonists and victims are "long tail" characters. Side characters would be long-tail characters, but antagonists are famously very prominent in stories.
   4. I wish authors had spent less time explaining why their new dataset is "the best" and instead focused on expanded discussion of previous literature, conceptual choices, etc. (see my other weaknesses).
6. The experimental setup, choice of baseline, and comparisons made:
   1. L269: For the sentence-level word lists (COMET or LIIPA), authors do not specify how they aggregate the word lists over sentences towards a full-story label.
   2. Furthermore, there is a conceptual issue here which is that sentence-level models might be accurate about predicting the portrayal at the sentence, but that does not necessarily reflect the portrayal of the character at the full story level. This issue is also related to the drop in performance for story-level character portrayals, which exhibit more of a drop with increased nb of sentences.
7. Other missing details and issues:
   1. Table 4: how did authors come up with the titles and genres?
   2. L186: Narrative quality is much more than diversity. Authors should either add more measures of quality (e.g., narrative flow, plot amount, etc.) or just say they measure diversity without mentioning quality.
   3. In Figure 3, it is not specified how authors extract the character roles for stories, making it hard to interpret the results faithfully.
   4. L261: COMET is a commonsense engine powered by an LLM (specifically, an LLM finetuned to perform commonsense inference). Saying that LLMs would outperform COMET doesn't make much sense if you don't specify which LLM powers COMET. I would rather say that a finetuned LLM for commonsense inference might not be as flexible as an out-of-the-box prompted LLM to extract these portrayals?
   5. L312: Authors should re-highlight that accuracy is computed based on the character attributes that were selected during generation.
   6. L322-328: I appreciate the discussion of result differences for different dimensions and labels. However, authors should also note that many modern LLMs' safeguarding might explain the refusal to categorize appearance, and their RLHF-induced over-agreeableness might induce their better accuracy for positive ("high") labels.
   7. L353: Authors mention correlations between performance and nb of characters or sentences. Authors should actually run these correlations and significance tests, because the slopes do not look all that different to me.

I hope the authors understand that I really do like this task, but there are many things that they could improve to make this paper more publication ready. Furthermore, given the subject of this work, I would actually suggest submitting to ACL Rolling Review to publish at NLP conferences, which tend to have more work around narrative analyses and would find a more excited audience there.

**Questions:**

NA

---

### Official Review · Reviewer_f7DK · 2024-11-03

**Soundness:** 3
**Presentation:** 3
**Contribution:** 2
**Rating:** 5
**Confidence:** 3

**Summary:**

This work introduces a new dataset and a set of prompts. The main aim is to study implicit character portrayals in text. Authors create a new dataset using LLMs that contain stories that they use for “uncovering implicit character portrayal”. The paper is well structured: authors introduce a new dataset, compare it with past datasets, proceed to use it and finally provide some insights (also some insights about fairness)

**Strengths:**

I find this paper interesting. The dataset is definitely the strongest contribution here and I can see this used to evaluate bias in LLMs. I also appreciate the depth of the evaluation, with different patterns that have been analyzed. I appreciate the structure of the appendix (and the table of contents!)

**Weaknesses:**

While I appreciate the dataset, I have an issue - it's LLM-generated and I'm not 100% confident about how general and representative it is. Looking at the examples, they are very "LLM-ish". Happy to hear the authors' opinion on this.

I noticed some issues with the implicit descriptions:

“Protagonist0, a well-meaning but **bumbling** individual, decided to plan a surprise party for their friend Victim0, who was known for their **good looks and charm**. Antagonist0, a **cunning and highly intelligent**”
In line 155 you mention you ask the llm to avoid explicit words like these - am I looking at the wrong thing here?

Also, in "The Awkward Date" example, I feel like the protagonist doesn't show high intellect, power, or appearance. Am I misunderstanding how to read the dataset?

The framework really needs a drawing, since there are several components involved and I feel like it is confusing. This could be added in place of Figure 1, which I think currently adds little to the paper.

**Questions:**

Some additional questions:
* Lines 278-311: This part was confusing for me, could you please describe this evaluation in more detail?
* I can’t find the results for the manual annotation, would you be able to share those?

---

### Official Review · Reviewer_kwww · 2024-11-03

**Soundness:** 2
**Presentation:** 3
**Contribution:** 2
**Rating:** 3
**Confidence:** 4

**Summary:**

This paper introduces ImPortPrompts (Implicit Portrayal Prompts), a dataset for implicit character portrayal analysis. The dataset is synthetically generated by GPT.  Furthermore, the authors  propose  LIIPA (LLMs for Inferring Implicit Portrayal for character Analysis), a framework for prompting LLMs to uncover character portrayals and investigate  fairness implications of synthetic dataset generation about characters and identify that various configurations
(structured output, prompt types) of LIIPA realize a fairness-accuracy tradeoff (with LIIPA-
DIRECT maximizing accuracy and LIIPA-SENTENCE minimizing unfairness). LIIPA-SENTENCE
and LIIPA-STORY outperform the prior baselines in terms of accuracy and fairness.

**Strengths:**

- The synthetically generated dataset, ImportPrompts, might be useful for studying characters in stories.

- Fairness analysis exposes fundamental biases in LLMs (e.g., when the protagonist is a man vs woman).

- The proposed prompts and methods are useful tools for analyzing character potrayal in stories

**Weaknesses:**

- The correctness of LLMs is nowhere assessed. LLMs are used for generating stories, for performing the character analysis and for assessing how well said analysis is fairing.

- The aim of this work is not entirely clear. If you wish to develop a tool for analyzing character portrayal in fiction, you ought to assess how this tool performs in *real* stories.

-Or can your method be used to correct LLM output?

- Please articulate clearly what your contributions are. You mention these on page 2, but creating a synthetic dataset without quality assessment is not a contribution. Is the LLIPA method novel? It seems similar to COMET on a conceptual level, but you are not using a knowledge base.

- The work is conducted with proprietary LLMs and this is not reproducible. Also it is not clear whether it extends to smaller, open-weights models.

**Questions:**

- Have evaluated the quality of your synthetic dataset? You mention that you have done a human evaluation in the Appendix, what are the actual numbers? How much noise is there in the dataset, how often are the instructions followed, and more importantly how good are the stories and how implicit is the portrayal fo the characters.

- Why do you think a story which is 20 sentences long is enough to analyze implicit character attributes? To genuinely understand and depict characters, works of fiction are much longer. Have you tried to see how your tool fares on human-authored stories? Is it human stories you are ultimately interested in or machine generated ones?

- What are the three dimensions of Intelect, appearance and power based on? Is there prior work in narrative theory claiming these dimensions are important? Also please tell us what is the avg length of stories in your dataset, and how many these are in total?

- Figure 4, are differences between depicted models statistically significant?

- You use a LLM-as-a-judge to assess the generated character attribute wordlists. This would made sense to me only if you show that the LLM correlates substantially with *human* quality judgments.

- Please show some examples of the attribute wordlists you generate and also example stories for each genre.

- I am puzzled as to why you think appearance can be described implicitly in a story, and how might this look?

---

### Note · Authors · 2024-11-27

**Comment:**

We thank the reviewers for their thorough and insightful comments which we plan to integrate in a future version of this paper.

**Withdrawal Confirmation:**

I have read and agree with the venue's withdrawal policy on behalf of myself and my co-authors.